# Kaposi’s Sarcoma in Virally Suppressed People Living with HIV: An Emerging Condition

**DOI:** 10.3390/cancers13225702

**Published:** 2021-11-15

**Authors:** Romain Palich, Alain Makinson, Marianne Veyri, Amélie Guihot, Marc-Antoine Valantin, Sylvie Brégigeon-Ronot, Isabelle Poizot-Martin, Caroline Solas, Sophie Grabar, Guillaume Martin-Blondel, Jean-Philippe Spano

**Affiliations:** 1Department of Infectious Diseases, Pitié-Salpêtrière Hospital, AP-HP, Pierre Louis Epidemiology and Public Health Institute (iPLESP), INSERM U1136, Sorbonne University, 75013 Paris, France; marc-antoine.valantin@aphp.fr; 2Infectious Diseases Department, INSERM U1175, University Hospital of Montpellier, 34000 Montpellier, France; a-makinson@chu-montpellier.fr; 3Department of Medical Oncology, Pitié Salpêtrière Hospital, AP-HP, Institut Universitaire de Cancérologie (IUC), CLIP² Galilée, Pierre Louis Epidemiology and Public Health Institute (iPLESP), INSERM U1136, Sorbonne University, 75013 Paris, France; marianne.veyri@aphp.fr (M.V.); jean-philippe.spano@aphp.fr (J.-P.S.); 4Department of Immunology, Pitié-Salpêtrière Hospital, AP-HP, Centre d’Immunologie et des Maladies Infectieuses, INSERM U1135, Sorbonne University, 75013 Paris, France; amelie.guihot@aphp.fr; 5Service d’Immuno-hématologie Clinique, Sciences Economiques & Sociales de la Santé & Traitement de l’Information Médicale, AP-HM, INSERM, IRD, Sciences Economiques et Sociales de la Santé et Traitement de l’Information Médicale, Hôpital Sainte-Marguerite, Aix Marseille University, 13007 Marseille, France; sylvie.ronot@ap-hm.fr (S.B.-R.); isabelle.poizot@ap-hm.fr (I.P.-M.); 6Department of Pharmacology Toxicology, AP-HM, Centre de Recherche en Cancérologie de Marseille, Hospital de la Timone, INSERM, Aix-Marseille University, 13005 Marseille, France; caroline.solas@ap-hm.fr; 7Pierre Louis Epidemiology and Public Health Institute (iPLESP), INSERM U1136, Sorbonne University, 75013 Paris, France; sophie.grabar@aphp.fr; 8Department of Infectious and Tropical Diseases, Centre de Physiopathologie Toulouse-Purpan, Toulouse University Hospital, INSERM U1043, CNRS UMR 5282, 31300 Toulouse, France; martin-blondel.g@chu-toulouse.fr

**Keywords:** Kaposi’s sarcoma, HIV, AIDS, antiretroviral, cancer

## Abstract

**Simple Summary:**

Kaposi’s sarcoma (KS) in people living with HIV (PLHIV) occurs in the vast majority of cases when viral replication is not controlled and when CD4 immunosuppression is important. However, clinicians are observing more and more cases of KS in PLHIV with suppressed viremia on antiretroviral treatment. These clinical forms seem less aggressive, but cause therapeutic dead ends. Indeed, despite repeated chemotherapy, recurrences are frequent. Immunotherapy and specific treatment regimens should be evaluated in this population.

**Abstract:**

Since the advent of highly effective combined antiretroviral treatment (cART), and with the implementation of large HIV testing programs and universal access to cART, the burden of AIDS-related comorbidities has dramatically decreased over time. The incidence of Kaposi’s sarcoma (SK), strongly associated with HIV replication and CD4 immunosuppression, was greatly reduced. However, KS remains the most common cancer in patients living with HIV (PLHIV). HIV physicians are increasingly faced with KS in virally suppressed HIV-patients, as reflected by increasing description of case series. Though SK seem less aggressive than those in PLHIV with uncontrolled HIV-disease, some may require systemic chemotherapy. Persistent lack of specific anti-HHV-8 cellular immunity could be involved in the physiopathology of these KS. These clinical forms are a real therapeutic challenge without possible short-term improvement of anti-HHV-8 immunity, and no active replication of HIV to control. The cumulative toxicity of chemotherapies repeatedly leads to a therapeutic dead end. The introduction or maintenance of protease inhibitors in cART does not seem to have an impact on the evolution of these KS. Research programs in this emerging condition are important to consider new strategies.

## 1. Introduction

Thanks to the efficacy and wide access of combined antiretroviral treatment (cART) for controlling HIV replication, the risk of Kaposi’s sarcoma (KS) in people living with HIV (PLHIV) has greatly declined over the past 25 years in resource rich settings [1,2,3,4]. However, KS still remains, with non-Hodgkin lymphoma (NHL) the most common cancer among PLHIV [2,3]. KS frequently occurs in uncontrolled HIV infection, but there are increasing descriptions of KS occurring in virally suppressed patients, even in those with apparent T CD4 immune restoration. In virally suppressed patients, KS may present as a de novo complication, or as a recurrent disease, with repeated episodes over time.

There are very few published case series of virologically controlled patients presenting with KS (Table 1) [5,6,7,8]. There is also no standardized definition of cases in terms of HIV-RNA duration of suppression and HIV-RNA and CD4 thresholds, hindering comparability between studies. Our group assumes that KS in virally suppressed patients is an emerging complication in PLHIV, with specific clinical, physiopathological and therapeutic difficulties, justifying a specific review to encourage further research. Thus, we aim to provide an overview of KS in virally suppressed patients on the (1) latest available epidemiological data, (2) reported clinical features, (3) immunopathological pathways, (4) therapeutic issues, including the impact of protease inhibitors.

## 2. Epidemiology of KS in Virally Suppressed HIV-Patients

Few observational cohorts have specifically studied the risk of KS in PLHIV with suppressed viremia. In the French ANRS CO4 FHDH cohort, PLHIV with undetectable plasma HIV-RNA and restored immunity (i.e., CD4 ≥ 500/mm^3^ for at least 2 years) still had a 35-fold higher risk of KS compared to the general population (standardized incidence ratio (SIR) = 35.4; 95% CI 18.3–61.9) [3]. Interestingly, in the same population, this high risk was in contrast to the NHL risk, another virus-related AIDS defining cancer (SIR = 1; 95% CI 4.0–1.8), suggesting a specific susceptibility to KS in PLHIV despite apparent immune and virological control. However, despite a follow-up of 55,633 person-years, observed events of KS were rare (*n* = 12) in PLHIV with controlled disease. More recently, the US Veteran cohort examined the risk of cancers from 1999 to 2015, and found that the risk of KS as compared to uninfected patients was still more than 50-fold in PLHIV despite long-term viral suppression (≥2 years) and more than 500-fold in those with early suppression (<2 years) [9]. 

KS has traditionally been described as occurring at low CD4 levels but the clinical context in which KS can occur have dramatically changed over the years and have impacted the pattern of presentation of KS. In the large collaboration of European cohorts, based on 1323 KS occurring at CD4 count ≥200/mm^3^, the incidence rate was 1.2 per 1000 (95%CI 1.1–1.2) and gradually decreased with increasing CD4 levels [10]. However, an increasing number of studies have reported greater proportion of KS occurring at higher CD4 cell counts [1,11,12] and some case studies reported KS occurring in aviremic patients (Table 1). In a study gathering data from 8 American cohorts, the authors showed that across 1996–2011 KS occurred at higher CD4 levels and lower VL [12]. Between 2007 and 2011, 15% of the KS occurred at CD4 count ≥500/mm^3^ and less than one half <200/mm^3^. They also showed that this trend was mainly explained by the increasing proportion of the underlying HIV population on effective cART who exhibited higher CD4 and suppressed plasma HIV-RNA, and not by an increased risk of KS within each stratum. Taken together, epidemiological data describes rare but possible KS occurrence in PLHIV with immune restoration and virological control, though numbers remain low. However, the increasing number of publications describing these cases underscore a potential increase of this condition with the aging of PLHIV. 

## 3. Clinical Presentation of KS in Virally Suppressed HIV-Patients

Data on disease severity are scarce, and probably biased by reporting of most severe cases. However, a 2006 report of nine PLHIV with controlled viremia (<300 copies/mL) and a sustained CD4 count ≥300 cells/mm^3^ reported indolent KS cases, as no eruptive cutaneous lesions nor visceral involvement or other AIDS defining illnesses were described [5]. In contrast, a retrospective study from the French CancerVIH group of 21 PLHIV and KS with a median viral suppression of 3 years (IQR 2-5), and a median CD4 level count of 449/mm^3^ reported frequent severe disease [7]. Eight PLHIV experienced a first episode of KS, all had skin lesion, six (27%) had lymph node involvement and eleven had visceral invasion (bronchial, bone and/or gastric lesions). These cases, referred to an expert national panel for cancer treatment advise for PLHIV, probably represented the more severe cases of KS in aviremic HIV-patients in France at that time. A monocentric study from France described all consecutive diagnosed cases of KS in aviremic (12 cases) and viremic (97 cases) HIV-patients diagnosed in a tertiary referral hospital between 2000 and 2017, comparing their clinical presentations with classic KS (also named “Mediterranean KS”, in HIV-negative individuals) consecutively diagnosed in the same area and extracted from the Francim cancer databank [8]. Locally skin indolent presentation was the main clinical presentation in 10 of the KS in aviremic patients, although one patient had visceral involvement and one a disseminated mucocutaneous form. KS in aviremic patients had similar semiology than the 62 classic KS, and, as expected, significantly more indolent presentations than KS in viremic patients. Thus, taking together, and as summarized in Table 2, most cases of aviremic may be more commonly indolent, though aggressive forms are not exceptional.

There is no classification for KS risk evaluation specific to HIV-patients with suppressed viremia. Thus, classifications used for KS in viremic patients may not be adequate in aviremic patients, and studies should determine whether these classifications predict survival, and/or necessity of systemic therapy in this population. Classifications of AIDS-related KS severity emerged with the AIDS epidemic. The Krigel score—from an initial clinical description of 49 men who have sex with men with KS—proposed four severity stages: locally indolent KS cutaneous lesions (stage I); locally invasive and aggressive form (stage II); disseminated mucocutaneous form (stage III), often with lymph node involvement; and disseminated, mucocutaneous form with visceral involvement (stage IV), further subtyped according to systemic signs of unexplained fever and/or weight loss [13]. There is no validation of this score, and clinicians and researchers felt that the four-stage classification systematically assigned most AIDS-associated cases to stage III or IV, and poorly predicted clinical outcomes [14]. Later, the classification developed by the AIDS Clinical Trial Group (ACTG) of the National Institute of Allergy and Infectious Diseases scored tumor localization and semiology (T), immune deficiency (I), and systemic illness (S) (i.e., fever) to classify subjects in poor or good risk groups [14]. Its predictive value was confirmed in a pre- cART cohort of 294 consecutive patients enrolled in eight ACTG therapeutic trials, with overall survival significantly shorter for patients in the poor-risk categories [15]. This classification remained in use after the advent of cART era. A study from the Swiss HIV cohort showed that staging T1 and a CD4 level <200 cells/mm^3^ were correlated with death (hazard ratio: 5.22 and 2.33, respectively) in 144 patients KS patients recruited between January 1996 and December 2004 [16]. The classification was also shown to be predictive of Immune Reconstitution Inflammatory Syndrome (IRIS)-associated KS (IRIS-KS) [17]. However, its prognosis value in PLHIV with controlled viremia and KS is unknown.

As KS in aviremic patients may be more commonly indolent [5,8], the use of alternative classifications from non-HIV populations is appealing. A proposed staging system derived from 300 CKS patients based on disease progression was developed [18]: stage I defined by small macules and nodules primarily confined to the lower extremities, stage II by infiltrative plaques mainly involving the lower extremities, sometimes associated with a few nodules, stage III with florid multiple angiomatous plaques and nodules involving the lower extremities that are often ulcerated, and a stage IV with disseminated disease. However, this classification is prone to subjective clinical assessment, and its prognosis value has not been independently validated in CKS studies. An approach based on localized disease (limited cutaneous disease) and advanced disease (advanced cutaneous, oral, visceral or nodal disease), as advocated by the National Comprehensive Cancer Network Guidelines (https://www.nccn.org/professionals/physician_gls/pdf/kaposi.pdf, accessed on 5 November 2021) are probably the most pragmatic guidelines to guide indications of therapeutic interventions, and could probably be also applied for KS in HIV-patients with controlled viremia despite the lack of clinical trials. According to these guidelines, limited cutaneous disease if symptomatic or cosmetically unacceptable is to be treated with topicals (alitretinoin 0.1% gel, imiquimod 5% cream), and for small lesions (i.e., ≤1 cm) intralesional chemotherapy (i.e., vinblastine), radiation therapy, local excision, or cryotherapy. For progressive disease, in case advanced cutaneous, oral, visceral, or nodal disease, first line systemic therapy with liposomal doxorobucin or paclitaxel are preferred options (please refer to guidelines for further details on indications and doses, and below for further discussion on therapeutic options).

## 4. Physiopathological Hypothesizes

We postulate that the pathological presentation of KS in HIV-patients optimally treated by cART constitutes a specific pattern, due to HHV-8 chronic antigen exposure, immune modulation by viral proteins, and local immune exhaustion. Several immunopathological hypothesis could explain the persistence of HHV-8 related KS during optimally controlled HIV infection with correct immune restoration, implicating persistent immune activation linked to HIV.

Alterations of the innate immunity and inflammation are strongly associated with KS [19], but the activation/inflammation status has not been described in KS patients with controlled plasmatic HIV-RNA so far. However, a decrease in CD4/CD8 ratio has been observed in these patients [20], suggesting that immune activation persists in periphery in this context. Indeed, a low CD4/CD8 ratio has been linked to T cell activation [21]. In vitro, various cytokines such as IL-1, IL-6, TNF-alpha and also Interferon-gamma induce proliferation of HHV-8 infected spindle cells [22]. Concordantly, basic fibroblast growth factor (bFGF) and HIV-1 Tat protein synergize in inducing angiogenic KS-like lesions in mice. Besides, the bFGF, extracellular Tat and Tat receptors are present in HIV-associated KS [23]. 

Other data suggest a loss of immune control by NK cells and T cells during HIV-related KS. Indeed, NK-cells have been reported to play a role in the anti-HHV-8 immune response, notably because HHV-8 down-regulates the MHC class I molecules expression on the HHV-8-infected tumor cells. Moreover, the strong expression of NKG2D ligands by tumor cells suggests a defect in NK-cell homing or survival in the KS microenvironment. Furthermore, NK-cells show functional exhaustion in KS patients with lack of response following direct triggering of NKp30, NKp46 or CD16 activating receptors [24]. Another study confirmed this hypo functional profile of NK cells in HIV-infected patients [19]. NK cells were not present in the tumor, suggesting a defect in NK cell cytotoxicity to tumor cells in the context of KS. 

Specific T cell responses to HHV-8 were also studied during HIV-related KS. We and others have shown that peripheral blood HHV-8-specific CD8 T-cell responses were much lower in HIV-infected patients with KS than in asymptomatic HHV-8-positive patients, regardless plasma HIV-RNA level [25,26]. Several hypotheses can be made to explain this defect. First HHV-8 encodes two gene products, K3 and K5 (also termed MIR1 and MIR2, respectively), which act in concert to efficiently downregulate the expression of MHC class I molecule on the surface of infected cells, thus preventing antiviral CTL responses [27]. Second, because KS lesions express PD-L1 [28], it is possible that T cells against HHV-8 undergo immune exhaustion. To date, data are not robust enough to conclude about the efficacy of anti-immune checkpoints for treating HIV-related KS. Finally, another hypothesis to the lack of T cell responses in peripheral blood would be the migration of these cells to KS tissues: this hypothesis has not been confirmed so far [25]. Taken together, data suggest that immunodominant HHV-8 proteins are probably not appropriately presented to the immune system during KS, and that T cell exhaustion could also be at play. 

## 5. Therapeutic Challenges for Treating KS in Virally Suppressed HIV-Patients

KS in virally suppressed PLHIV is challenging, as control of HIV replication after introduction of cART and immune improvement are no longer part of the strategy [29].

Since the mid-1990s, highly effective cART has been available, which allow optimal suppression of viral replication. Still currently, cART is based in the vast majority of cases on a combination of two NRTIs and a third agent, to be chosen from among PIs, NNRTIs or INSTIs. From 1996, PIs and NNRTIs were available, but INSTIs were not marketed until the end of the 2000s. The four series of virally suppressed PLHIV with KS we report in this article (Table 1) include patients treated in the last 10–15 years, and during this period, antiretroviral drugs have changed. In particular, INSTIs now occupy a central position. There is no data in the literature supporting the impact of this change on the incidence of KS. However, the protective role of PIs has been the subject of several works, reported below in this review.

In patients already on cART, KS—once the options of surveillance and local treatment are lapsed—will inevitably require systemic chemotherapy, such as anthracyclines or taxanes (Table 3) [30]. The difference in the proportion of patients having received chemotherapy in the different series reported here could be explained by more or less severe clinical presentations. If recurrences occur despite initial systemic chemotherapy, reiterative cure with inherent risk of cumulative toxicities commonly leads to KS treatment dead ends. As an example, the maximum cumulative dose of liposomal anthracycline is limited to ~550 mg/m^2^ to preserve cardiac functionality, under regular cardiac monitoring [31]. For paclitaxel, it is not uncommon for peripheral neuropathies to prevent continued treatment after a high number of perfusions [32].

Alternative therapies exist, but none are fully satisfactory or validated, and further clinical trials are warranted. Moreover, all strategies have been studies in KS in PLHIV with uncontrolled viral replication. Bleomycin monotherapy can be used, as a suboptimal option, this chemotherapy being less effective than anthracyclines and taxanes for AIDS-related KD, with a risk of pulmonary toxicity [33]. Beside cytotoxic drugs, other treatments have been evaluated in small studies, such as antivascular endothelial growth factors (lenalidomide and pomalidomide [34,35]) and peginterferon alfa-2a [36], but these are limited by toxicities, limited efficacy. Moreover, most of these chemotherapies present a high potential risk of a drug–drug interaction with cART. In a limited number of patients, topical treatments may also be offered, such as radiotherapy, cryotherapy or topical retinoids, but often systemic treatment is required in case of recurrence despite these treatments.

The management of these patients is based on the general guidelines for AIDS-associated KS, while the therapeutic needs and tolerance are undoubtedly different. The cases reported in the literature support more indolent forms in virally suppressed patients, with, probably, visceral involvement which are rarely life-threatening. Chemotherapies with lower doses could be discussed, as well as a spacing of cures, for better tolerability over time. Induction treatment with chemotherapy, followed by maintenance or suspensive treatment with oral drugs to avoid recurrences could be evaluated, However, drug candidates for maintenance treatment remain to be defined.

Immunotherapy could be an interesting therapeutic option in patients with AIDS-related KS despite suppressed viremia on cART. Immunotherapy relies on monoclonal antibodies blocking immune checkpoints, such as ipilimumab, blocking CTLA-4, nivolumab and pembrolizumab, blocking PD-1, or atezolizumab, blocking PD-L1. The lack of specific anti-HHV-8 cellular immunity has been shown to participate in the occurrence of KS [26]. Blocking immune checkpoints could restore anti-HHV-8 immunity and help control the tumor process. The expression of PD-1 and PD-L1 within tumor tissue has been shown to be a marker strongly correlated with the effectiveness of immunotherapy [37]. Some histological works show the expression of these immune checkpoints from KS biopsies [28,38,39,40,41,42], supporting possible therapeutic effects of immune checkpoints inhibitors on KS. Only one observational study reported AIDS-related KS treated by anti-PD-1 therapy [43]. Among the 9 reported cases, 6 were on cART for at least 12 months, with HIV-RNA <50 copies/mL. All had received previous treatments, included liposomal anthracycline, paclitaxel, lenalidomide and bortezomib. All had cutaneous involvement, with lymph node (*n* = 3), gastro-intestinal (*n* = 2) and lung (*n* = 1) extension. The authors concluded that there was partial remission in 4 patients, and stable disease in 2 patients, with a median follow-up of 5 months. Regarding adverse events, several cohorts have shown similar tolerance of immunotherapy in PLHIV than in people without HIV [44,45,46,47]. In particular, no immunological and virological effects were detected in these patients. Immune-related adverse events are commonly mild to moderate, included skin, musculoskeletal, gastrointestinal and endocrine impairment. However, some rare and serious side effects can occur with a real risk of death, including immune-induced pneumonia or cardiomyopathy. Overall, although there is no marketing approval for anti-immune checkpoint antibodies for KS treatment today, this therapeutic way could be an interesting option for PLHIV with suppressed viremia. Nevertheless, indications for immunotherapy should be balanced against the risk of serious adverse reactions, especially when KS does not endanger the patient’s life.

## 6. Role of Protease Inhibitors for Treatment

HIV PIs were shown in vitro to have through their action on cellular proteases a wide range of effects on pathways that are important for tumorigenesis, including reducing angiogenesis and cell invasion, inhibition of the Akt pathway, induction of autophagy, and promotion of apoptosis [48]. These properties, in addition but independently of the effect of PIs on inhibition of the HIV protease and HIV replication, were suggested in vitro and in experimental models to be potentially beneficial on KS prevention and management. 

The first-generation PIs ritonavir and saquinavir inhibit in vitro activation and proliferation of primary endothelial cells and KS cell lines through induction of apoptosis of tumor cells by modulating proteasomal proteolysis without affecting proliferation or survival of noncancerous cell. They also decrease production of vascular endothelial growth factor (VEGF) and inflammatory cytokines (tumor necrosis factor-α, interleukin-6 and -8), which are critical to KS development and proliferation [49,50]. Ritonavir inhibited tumor formation and progression by KS-derived cells in a KS mouse xenotransplantation model [50]. Systemic administration of indinavir and saquinavir to nude mice blocked the development and induced regression of angioproliferative KS-like lesions established by primary human KS cells by the inhibition of matrix metalloproteinase-2 proteolytic activation at concentrations present in plasma of treated individuals [51]. Moreover, nelfinavir, but no other PIs, was shown to be a potent inhibitor of HHV-8 replication in vitro through interference with the production of infectious virus [52]. In a small uncontrolled study that aimed to assess the effectiveness of indinavir as a therapy for classical KS in HIV-seronegative patients, favorable clinical outcome was achieved in 16/26 patients (61.5%), and was associated with higher plasmatic indinavir concentrations, reduced plasmatic levels of basic fibroblast growth factor, lower numbers of circulating endothelial cells, and decreased antibody titers against HHV-8, compared to patients with unfavorable course, although HHV-8 viral load was not monitored [53]. These data were followed by case series and case reports describing onset or relapses of KS despite long-term remission under PI-based regimen after being switched to a non-nucleoside reverse transcriptase inhibitor (NNRTI)- or integrase strand transfer inhibitor (INSTI)-based ART [54,55,56]. A retrospective study performed on the Veterans Affairs HIV Clinical Case Registry showed that longer duration on ritonavir boosted PI-based regimen significantly reduced KS incidence among male Veterans, in comparison with other regimens, after accounting for potential confounders including HIV viral load, CD4 T cell count and IRIS effect, with a most pronounced protective effect after at least one year on ART. This effect was not demonstrated for nelfinavir, non-boosted PIs and NNRTI-based regimen [57]. 

However, the potential effect of PIs on KS has not obviously translated into the clinic. Onset of KS was reported in patients with controlled HIV replication while on PI-based ART regimen [5,58]. Several cohorts (EuroSIDA, COHERE in EuroCoord, French Hospital Database on HIV, Chelsea and Westminster HIV cohort) assessing incidence and risk factors for KS stated that boosted PI-based regimens were not associated with a lower risk of developing KS than NNRTI-based regimens in the overall population or in men who have sex with men [1,10,59,60,61]. Of note these studies usually did not adjust for possible KS-IRIS effect after ART initiation representing a potential confounding factor in the interpretation of their results [62]. In cohorts of KS patients that aimed to identify predictive factors of KS remission, response rate was not associated with NNRTI-based, PI-based, or boosted-PI based regimen [63,64]. Furthermore, an analysis of the prospectively collected Dat’AIDS database focusing on PLHIV with history of KS and controlled HIV replication, switching from a PI-based to a PI-free regimen was not associated with an increased risk of KS relapse [65].

Therefore, while being a relevant issue in the setting of current recommendations of ART optimization with PI-free regimens [66], and despite a tempting pre-clinical rationale, no clinical data supports a beneficial direct effect of PIs on HHV-8 and KS. Both PI- and NNRTI-based ART were shown to be equally effective in protecting against KS. This suggest that reduction in KS incidence on ART is related to improved immune recovery and HIV control, and that KS relapses in the setting of controlled HIV replication could be mainly mediated by permanent loss of the anti-HHV-8 T cell immune responses despite CD4 T cell quantitative restoration under ART [26], rather than by a specific anti-HHV-8 or antiangiogenic effect of PI-containing regimen.

## 7. Conclusions

KS in virally suppressed HIV-patients represents a clinical and biological entity that is still poorly understood. Exploration of immunopathological pathways is important to develop more effective therapeutic strategies, by avoiding reiterative cytotoxic chemotherapy. The search for genetic abnormalities by high-throughput sequencing of tumor tissue could also reveal previously unidentified therapeutic targets. We believe that it is necessary to constitute a prospective cohort including HIV-patients with KS despite suppressed HIV viremia on cART, in order to collect exhaustive data on these cases, and to describe the evolution of the disease according to the treatments received, with their potential associated adverse effects, and factors associated with negative outcomes. Such a cohort is now ongoing on all patients presented at the expert panel. We plead also for the implementation of clinical trials specifically enrolling virally suppressed PLHIV, in order to evaluate innovative approaches, such as induction/maintenance schemes, based on drug-reduces cytotoxic chemotherapy, immunotherapy and oral new targeted therapies.

## Figures and Tables

**Table 1 cancers-13-05702-t001:** Main published series including Kaposi’s sarcoma in HIV-patients with suppressed viremia.

Reference	Country	Number of Subjects	HIV-RNA Value Restrictions for Cases to Be Included	CD4 Count Restrictions for Cases to Be Included	Median CD4 Count	Median CD4 Nadir	cART Conditions to Be Included	Median Duration on cART	Types of cART
[5]	USA	9	<300 copies/mL for at least 2 years	≥300/mm^3^	-	340/mm^3^ (range: 90–455)	-	7 years (range: <1–19)	PI-based or NNRTI-based therapy
[6]	USA	20	<75 c opies/mL	≥300/mm^3^	483/mm^3^ (range: 300–625)	216/mm^3^ (range: 4–431)	-	5 years (range: 1–12)	PI-based (*n* = 11, 55%) or NNRTI-based (*n* = 9, 45%) therapy
[7]	France	21	<50 copies/mL	-	449/mm^3^ (IQR: 241–625)	196/mm^3^ (IQR: 84–329)	≥12 months	-	PI-based (*n* = 4, 19%) or NNRTI-based (*n* = 7, 33%) or INSTI-based (*n* = 10, 48%) therapy
[8]	France	12	<50 copies/mL for at least 12 months	-	723/mm^3^ (range: 520–881)	-	-	-	-

PI, protease inhibitor. NNRTI, non-nucleoside reverse transcriptase inhibitor. INSTI, integrase stand transfer inhibitor.

**Table 2 cancers-13-05702-t002:** Clinical presentation of Kaposi’s sarcoma in HIV-patients with suppressed viremia in main published series.

Reference	Number of Subjects: Total (Male/Female)	Median Age of Subjects: Years	Skin or Palatine Involvement	Lymph Node Involvement	Visceral Involvement Oedema or Ulceration or Nodular Oral Lesions
[5]	9 (-/-)	51 (range: 41–74)	9 (100%)	0	0
[6]	20 (19/1)	42 (range: 25–59)	12 (60%)		8 (40%)
[7]	21 (17/4)	54 (interquartile: 35–61)	Skin, 21 (100%) Palatine, 1 (5%)	6 (27%)	Bronchi, 4 (18%) Bone, 4 (18%) Stomach/esophagus, 3 (14%)
[8]	12 (12/0)	54 (range: 38–60)	Skin 10 (83.3%)	1 (8.3%)	Lung, 1 (8.3%)

**Table 3 cancers-13-05702-t003:** Treatments used for the management of Kaposi’s sarcoma in HIV-patients with suppressed viremia, in addition to combined antiretroviral therapy, and clinical outcomes, in the main published series.

Reference	Mani, *J Int Assoc Physicians AIDS Care*, 2009 [6] (*n* = 20)	Palich, *Clin Infect Dis*, 2019 [7] (*n* = 21)	Severin, *AIDS*, 2021 [8] (*n* = 12)
All cytotoxic treatments	14 (70%)	21 (100%)	2 (17%)
Liposomal doxorubicine	13 (65%)	19 (90%)	-
Paclitaxel	3 (15%)	10 (48%)	-
Bleomycin	-	2 (10%)	-
Vincristine	-	1 (5%)	-
Antivascular endothelial growth factors	-	5 (24%)	-
Interferon-alfa	-	2 (10%)	-
All local treatments	4 (20%)	1 (5%)	4 (33%)
Radiotherapy	4 (20%)	1 (5%)	1 (8%)
Cryotherapy	-	-	1 (8%)
Retinoid	-	-	1 (8%)
Laser	-	-	1 (8%)
Unspecified other treatment	5 (25%)	-	-
Clinical surveillance only	1 (5%)	-	5 (42%)
Clinical outcomes			
Follow-up duration, months, median	39 (range: 6–120)	17 (interquartile: 9–20)	-
Complete or partial regression	13/20 (65%)	6/16 (37%)	-
Stable disease	4/20 (20%)	6/16 (38%)	-
Progression	3/20 (15%)	4/16 (25%)	-

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
