# Peer review of "Kaposi’s Sarcoma in Virally Suppressed People Living with HIV: An Emerging Condition"

_cancers, 2021, doi:10.3390/cancers13225702_

Round 1

Reviewer 1 Report

Palich et al. have done an overall great job of describing the new challenges that clinicians face in treatment of PLHIV. The rise of KS cases in PLHIV with suppressed viremia recently has created an unmet need for a body of literature and promotion for further investigation.

However, there are a few points that could add more value to the current manuscript:

  1. What do authors mean by the "Series of virologically controlled patients presenting with KS are seldom” ?
  2. It would increase the value of the findings and conclusions if TABLE.1 is modified to include more demographic data of participants in the 4 main studies used for this manuscript specially gender and median age as KSHV represents a gender bias that has not been well understood, and the current manuscript can shine a light on that. (Reference: Dittmer DP, Damania B. Kaposi's Sarcoma-Associated Herpesvirus (KSHV)-Associated Disease in the AIDS Patient: An Update. Cancer Treat Res. 2019;177:63-80. doi:10.1007/978-3-030-03502-0_3).
  3. The Authors have investigated the link between the outcomes and immune reconstitution inflammatory syndrome (IRIS)-associated KS it would also be very interesting to investigate symptoms of KICS (KSHV-associated inflammatory cytokine syndrome (KICS) as studies have shown viral suppressed patients with KS had been diagnosed positive with that.
  4. Have any of the patients developed MCD? Or presented with Anemia and hypoalbuminemia?
  5. Additionally, if there are any data available on interleukin levels it would be great to be added in the form of a table which would be of particular interest.

Reviewer 2 Report

Article Comments KS and ART Article

Brief Summary

Overall, I think this is a very well written and comprehensive review of Kaposi sarcoma (KS) among HIV virally suppressed adults. This study includes a thorough review of the topic and a detailed analysis of current literature and fills a gap in the current understanding about KS.  One of the strengths of the article is the excellent discussion about the therapeutic challenges to treat KS in virally suppressed patients with HIV and the call for future research to focus on improved therapeutics for these patients, specifically immunotherapy.  In summary, I think this manuscript is thorough and comprehensive and fills an important gap in the current KS literature.

General concept comments

As the therapeutic challenges for treating KS in virally suppressed patients with HIV is a significant focus of the manuscript, perhaps it would be helpful to comment in a bit more detail regarding the limitations of the ACTG classification and its use in determining a therapeutic approach for adults with KS. As the ACTG staging was developed in the pre-ART era perhaps it doesn’t well address the substantial heterogeneity of adults with KS, particularly among adults with KS who have HIV viral suppression.  The authors suggest utilizing an approach based on the National Comprehensive Cancer Care Network Guidelines to guide therapeutic interventions – a more detailed description of this approach and how therapeutic interventions could be selected if patients were classified with those guidelines could help to make the clinical presentation section (section 3) more complete.

As the published series include cohorts of patients from over the past 10-15 years, perhaps it would be helpful to comment on the role of changing available ART regimens over time – in particular the shifts from NNRTI based treatment to PI based treatment to INSTI based treatment and if the changes in types of ART patients receive have impacted KS clinical presentation. There is an excellent section describing the potential role of protease inhibitors already included in the manuscript, but it could be helpful to provide some historical perspective on the changing and improving ART regimens over the past 10-15 years.

Specific concept comments

Do the main published studies that are included in the manuscript mention the type of ART regimen the participants in the study were receiving?  This could be interesting to include, perhaps in Table 1.

Did the main published studies include information about the therapeutic approach utilized for their patients?  The wide variation in treatment approaches is interesting, as demonstrated by the percentages of patients who received cytotoxic treatments ie 100% of patients in the study by Palich compared with 17% of patients reported by Severin.  Is this variation due to variations in disease severity?  Or some other factor?

Are the outcomes known of the participants from these studies included?  As one of the main points is the therapeutic challenge of caring for patients with KS in the setting of HIV viral suppression, it would be interesting to know the outcomes of these patients and if they experienced treatment related toxicity.
